# Flaxseed Polysaccharide Alters Colonic Gene Expression of Lipid Metabolism and Energy Metabolism in Obese Rats

**DOI:** 10.3390/foods11131991

**Published:** 2022-07-05

**Authors:** Hua Wei, Xiaohong Lin, Liu Liu, Xichun Peng

**Affiliations:** 1College of Food Science and Engineering, Nanchang Univeristy, Nanchang 330096, China; weihua@ncu.edu.cn; 2Department of Food Science and Engineering, Jinan University, Guangzhou 510632, China; lxh781513747@163.com (X.L.); cdliuliu@126.com (L.L.)

**Keywords:** flaxseed polysaccharide, colonic gene expression, lipid metabolism

## Abstract

Obesity is one of the most serious public health challenges. Recently, we found that flaxseed polysaccharides (FPs) had an anti-obesity effect through promoting lipid metabolism, but the obesity-inhibiting pathway of FP is still unclear. In this study, after FP intervention in an obese rat model, a transcriptome study was performed to further investigate how FP intervention alters the gene expression of colonic epithelial tissues (CETs). The results showed that there were 3785 genes differentially expressed due to the FP intervention, namely 374 downregulated and 3411 upregulated genes. After analyzing all the differentially expressed genes, two classical KEGG pathways were found to be related to obesity, namely the PPAR-signaling pathway and energy metabolism, involving genes *Fabp1–5*, *Lpl*, *Gyk*, *Qqp7*, *Pparg*, *Rxrg*, *Acsl1*, *Acsl4*, *Acsl6*, *Cpt1c, Car1–4, Ca5b, Car8, Car12–14, Cps1, Ndufa4l2, Cox6b2, Atp6v1g2, Ndufa4l2* and *Cox4i2*. QRT-PCR results showed a consistent expression trend. Our results indicate that FP promotes lipid metabolism by changing the expression of some key genes of CETs, thus inhibiting obesity.

## 1. Introduction

Obesity is a public health challenge we are facing, the causes of which include biological, behavioral, and environmental factors [1]. According to the World Health Organization, more than 340 million children and adolescents aged 5–19 years were found to be overweight or obese in 2016 [2]. Developing from the disproportion between energy expenditure and caloric intake [3], obesity is associated with increased risk for diabetes, cancer, and other chronic diseases [4].

Elevated dietary fiber intake is effective at reducing obesity and metabolic syndrome induced by a high-fat diet [5]. Dietary fiber resists digestion in the small intestine, and passes into the large intestine where it may undergo partial or entire fermentation by the intestinal microflora [6]. It was reported that some dietary fibers influenced the genes of colonic epithelial cells that were involved in the pathways participating in substrate oxidation to produce energy [7]. Furthermore, gut microbiota can influence energy acquisition from dietary ingredients, and rats’ genes regulate energy expenditure and storage [8].

Flaxseed polysaccharide (FP) is an active complex carbohydrate (polysaccharide) from flaxseed shell [9]. As is known to us, polysaccharide with a complex structure cannot be degraded in the simulated digestive system, but decomposed and used by bacteria after integrally reaching the large intestine [10]. The intestinal bacteria will ferment or degrade the polysaccharide and produce short-chain fatty acids, oligosaccharides, etc. These products will pass through the intestinal mucus layer and interact with epithelial tissues, which leads to the various effects of polysaccharides [11]. 

FP is a heteropolysaccharide composed of acidic and neutral components [9]. In our previous research, we found that FP has an anti-obesity effect via promoting lipid metabolism, inducing satiety, and regulating the intestinal flora [12,13]. As a polysaccharide, FP cannot be digested and absorbed directly as nutrients by the body to play a role, so the obesity-inhibiting FP pathway is still unclear. We speculate that FP will be fermented or degraded by gut microbiota, and then alter the gene expression of CETs and regulate lipid metabolism and energy metabolism in vivo through its interaction with intestinal flora. However, it was unknown which gene expression in CETs is changed by FP. In this study, we aimed to further investigate the mechanism by which FP may induce weight loss by identifying genes of CETs that are differentially expressed in the presence and absence of dietary FP.

## 2. Materials and Methods

### 2.1. Chemicals and Reagents

TRIzol reagent was bought (Invitrogen, Carlsbad, CA, USA). A Bestar^TM^ qPCR RT kit was purchased (DBI Bioscience Co., Shanghai, China). Flaxseed polysaccharides (FPs) were generously provided by Professor Yong Wang at Jinan University. Briefly, the flaxseed shell was isolated from crushed flaxseed, and the crude polysaccharides were obtained after water extraction, alcohol precipitation and deproteinization. After ion-exchange column chromatography and dextran-gel column chromatography, FPs were prepared [14]. It contained two fractions, FP-1 and FP-2, with molecular weights of 2626 kDa and 1182 kDa, respectively, based on their analysis [14]. The FP used in this experiment contained these two components.

### 2.2. Animals, Diets, and Sample Preparation 

As shown in our previous research [13], eighteen male Sprague–Dawley (SPF grade) rats, 4 weeks old, were bought from Guangdong Medical Laboratory Animal Center (Guangzhou China). After 10 days of adaptive feeding with a standard diet (D12450B), the rats were randomly divided into two groups, namely the control group (Group Con, n = 6) and the obesity group (n = 12), to establish the obesity model. Rats in Group Con were fed with a standard diet (D12450B), and rats in the obesity group were given a high-fat diet (D12492). After about 8 weeks, the average weight of the rats in the obesity group (467.8 ± 19.9 g) was 20% higher than that of Group Con (384.6 ± 6.9 g), indicating the establishment of the obesity model. Subsequently, rats in the obesity group were randomly divided (n = 6 for each) into High-fat group (Group HFD) and FP-diet group (Group FPD). Group Con and Group HFD were fed the control diet (AIN-93M), and Group FPD was fed an FP diet (10% cornstarch in AIN-93M was replaced by the same amount of FP) for 52 days. The rats were weighed weekly. The detailed recipes of each diet are shown in Table 1.

On day 53, all rats were fasted overnight and were anesthetized with sodium pentobarbital. After the execution of the rats, the CETs were obtained with a scalpel and then placed on disinfected silver paper. They were frozen in liquid nitrogen immediately and stored at −80℃ until RNA extraction. The experiments were ratified by the Research Animal Administration Center of Jinan University (Guangzhou, China, Approval No.20180402-03), following all Institutional Animal Care and Use Committee of Jinan University guidelines for the use and care of animals.

### 2.3. cDNA Library Construction, RNA-seq and Bioinformatics Analysis

These processes were performed as described previously [15]. Briefly, total RNA of the colonic epithelial tissues was extracted using TRIzol reagent. The RNA quality was determined and quantified. The cDNA library was generated using high-quality RNA samples, and was sequenced by the Illumina HiSeq 4000. Raw read data from RNA-seq were filtered using SeqPrep and Sickle software. Clean reads were mapped to the reference genome with bowtie2 software. Gene abundances were quantified by RSEM. DEGs were determined using DEseq2. If the adjusted *p*-value < 0.05, the genes were considered to be significantly differentially expressed. Subsequently, KEGG Pathway analysis, KEGG Pathway enrichment analysis as well as PPI (protein-protein interaction) network analysis were performed [16]. 

### 2.4. Quantitative Real-Time PCR (qRT-PCR) Analysis

TRIzol reagents were used to extract total RNA. The cDNA was synthesized using a Bestar^TM^ qPCR RT kit and the gene mRNA expression was determined with a BestarTM SybrGreen qPCR Master Mix. The PCR conditions were maintained at 95 °C for 2 min, followed by 40 cycles of 94 °C for 20 s, 58 °C for 20 s and 72 °C for 20 s. The program Primer 5 (Premier, Walnut Creek, CA, USA) was used to design primers for PCR. The sequences of all the primers used are provided in Table 2. The average cycle threshold (Ct) values were used for quantification using the 2^−∆∆Ct^ method. *β-actin* was used as reference gene.

### 2.5. Statistical Analysis

Numerical data are expressed as means ± standard deviation (SD). SPSS 25.0 software (IBM Corporation, Armonk, NY, USA) was used to analyze the difference between the means. Two-tailed Student’s t-test was used to compare the data of different groups. *p* < 0.05 (*) or *p* < 0.01 (**) was considered statistically significant.

## 3. Results

### 3.1. Gene Expression Profile of CETs

After feeding FP for 54 days at the end of the trial, the weight of rats in Group FPD decreased by 9.81% compared with that in Group HFD (Figure 1), which indicated that FP induced body weight loss. An average of 52.16 million reads was obtained, and a large portion (95.14%) of high-quality clean reads were generated. The proportion of clean reads with a basic quality greater or equal to Q30 is higher than 95.14% with 48.37–50.25% GC content, which indicated the high quality of data. An average of 96.04% clean reads was mapped to the reference genome, among which 90.06% were mapped to the unique position. 

As shown in Figure 2a, 80.45% of the genes were in common among all three groups. Furthermore, the genes between Group Con and Group HFD were somewhat similar, while Group FPD was obviously different compared with them (Figure 2b). The result indicated that FP intervention would change the expression of CETs.

The variation in mRNA expression between Group Con and Group HFD is shown in Figure 2c. A total of 28 DEGs, including 14 upregulated and 14 downregulated, were identified. There were 4344 DEGs identified between Group Con and Group FPD (Figure 2d), including 476 downregulated and 3868 upregulated (*p* < 0.05). Furthermore, 3785 DEGs were identified between Group HFD and Group FPD (Figure 2e), including 374 downregulated and 3411 upregulated DEGs (*p* < 0.05). 

### 3.2. Analysis of DEGs Profile

To study the mechanism by which FP may induce weight loss in obese rats, we further analyzed the DEGs between Group HFD and Group FPD. The DEGs were mapped in the KEGG pathway database. In the comparison of the HFD and FPD treatments, annotated genes were divided into six categories. Most of them were enriched in signal transduction (Figure 3). It has been reported that weight loss involves a variety of mechanisms, such as inhibiting energy intake and stimulating energy expenditure. Furthermore, it has been found that the enrichment of colonic genes in the PPAR-signaling pathway was closely related to the anti-obesity effect [17]. In this research, 15 DEGs were found to be enriched in the energy-metabolism pathway and 26 DEGs in the PPAR-signaling pathway.

Through KEGG pathway enrichment analysis of DEGs in ‘energy metabolism’ and the ‘PPAR-signaling pathway’ between Group HFD and Group FPD, we further studied which biochemical reactions these genes participated in. The top 10 ranked KEGG pathways of DEGs in ‘energy metabolism’ are shown in Figure 4a. ‘Nitrogen metabolism’ had the highest rich factor and therefore showed the strongest enrichment degree, followed by ‘monobactam biosynthesis’. ‘Nitrogen metabolism’ showed the most DEGs, followed by ‘oxidative phosphorylation’, and the DEGs in the two pathways are listed in Table 3. Furthermore, Figure 4b shows the top 10 ranked KEGG pathways of DEGs in the PPAR-signaling pathway. The adipocytokine-signaling pathway showed the most DEGs, followed by ‘fatty acid degradation’ and ‘fatty acid biosynthesis’. The DEGs enriched in the PPAR-signaling pathway were further analyzed and the PPI network (Figure 5a) shows the associations between 23 DEGs (Table 3).

### 3.3. Verification of DEGs 

To verify the results of transcriptome sequencing, 17 important DEGs were selected for qRT-PCR, and the results are shown in Figure 5b. The expression trends were consistent with those obtained by RNA-seq, indicating that the RNA-seq data reliably reflected the change in gene expression.

## 4. Discussion

This experiment investigated the gene expression of CETs affected by FP intervention in an obese rat model. Compared with Group HFD and Group FPD, 3785 DEGs were found after FP intervention, 15 of which were enriched in the energy-metabolism pathway. Furthermore, 23 DEGs in the PPAR-signaling pathway showed strong connection.

### 4.1. FP Intervention Regulated Lipid Metabolism of CETs

The potential mechanism by which FP intervention may alter lipid metabolism of CETs is shown in Figure 6. In our study, we found that FP intervention altered the expression of 10 genes related to lipid metabolism, namely *Fabp1–5*, *Lpl*, *Gyk*, *Qqp7*, *Pparg*, *Rxrg*, *Acsl1*, *Acsl4*, *Acsl6* and *Cpt1c*. These genes play different roles in the process of lipid metabolism. 

FP intervention upregulated the expression of *Fabp1–5* and *Lpl*. FABPs mediate the transport of fatty acids (FAs) to different organelles [18]. Upon entry into intestinal cells, free FA and glycerol are metabolized or re-esterified into triglycerides (TG), which are secreted into the lymph after being packaged as chylomicrons [19]. The reduced clearance or increased yields of chylomicrons can lead to hypertriglyceridemia [20]. The processed Chylomicrons-TG can be metabolized at the tissue level through LPL, releasing FA for tissue uptake [21]. Thus, the upregulation of *Lpl* might inhibit the release of chylomicrons and the rise of plasma TG levels. In our previous study, it was found that the serum TG level of FP-fed rats was significantly lower than that of obese rats [13], which is consistent with our current inference.

FP intervention downregulated *Gyk* and upregulated *Aqp7*. Catalyzed by glycerol kinase, the product of glycerol, glycerol 3-phosphate, can be the main substrate in the synthesis of TG [22]. AQP7 facilitates the transport of glycerol across cell membranes [23]. Therefore, FP intervention might inhibit the synthesis of TG and facilitate the transport of glycerol that would be metabolized in other tissues. 

It is also very important to accelerate the oxidation of FAs, which in turn helps to inhibit their re-esterification. In the present study, the upregulation of *Pparg* and *Rxrg* was found. Several prior studies have demonstrated that PPAR agonists can induce the expression of lipid oxidation genes [24]. The heterodimeric partners of PPARs include RXR [25]. Researchers found that the activation of PPARγ could control the pathway related to FA metabolism [19]. Thus, FP intervention might alter the transcription of genes related to lipid metabolism by enhancing the activation of PPARγ and RXR.

FP intervention upregulated *Acsl1*, *Acsl4*, *Acsl6* and *Cpt1c*. ACSL catalyzes the form of acyl-coenzyme A (CoA), which enters the mitochondria matrix through the mediation of the CPT system [26,27]. Thus, FP intervention might promote the conversion of FA to acyl-CoA and the mitochondrial transport. FP intervention also upregulated *Plin1*, *Adipoq*, *Angptl4* and *Scd*, which were found to play an important role in the prevention and regulation of inflammation [28,29,30,31]. Our previous study showed that, compared with obese rats, the serum inflammatory cytokines, including IL-6, TNF-α, and IL-1 β, were significantly decreased in FP-fed rats, which is in accordance with the inference that FP intervention can inhibit inflammation [13].

In general, the regulation of lipid metabolism by FP intervention may be mainly through the regulation of the PPAR-signaling pathway, thus stimulating FA oxidation and reducing inflammation.

### 4.2. FP Intervention Regulated Energy Metabolism of CETs

The potential mechanism by which FP intervention may alter energy metabolism of CETs is shown in Figure 7. Many Carbonic anhydrases (CAs, including *Car1–4*, *Ca5b*, *Car8* and *Car12–14*) were found to be upregulated, which catalyze the reaction of water to form protons and bicarbonate (H_2_O + CO_2_

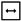
 H_2_CO_3_

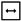
 HCO_3_^−^ + H+) [32]. Thus, FP intervention upregulated CAs and might accelerate the consumption of CO_2_, reducing its accumulation in cells and enhancing the tricarboxylic acid (TCA) cycle. This would further increase energy consumption. We also found that the upregulation of *Cps1*, which participated in the metabolism of metabolic ammonia and HCO_3_^−^ into citrulline with *Ca5b*, plays an important part in the detoxification of intestinal ammonia [33,34]. 

FP intervention upregulated *Ndufa4l2*, *Cox6b2* and *Atp6v1g2*. Generated from the TCA cycle and β-oxidation, NADH and FADH2 provide electrons to the electron transport chain [35]. NDUFA4L2 was found to fine-tune the activity of complex I, and COX6B2 promoted the assembly of complex IV [36,37]. V1 of ATP6V1G2 contains the sites of ATP hydrolysis [38]. Thus, FP intervention might help to accelerate the generation of ATP, stimulate energy consumption and hydrolyze ATP to provide energy for H+ transport.

FP intervention upregulated *Ndufa4l2* and *Cox4i2*. As the TCAs progress and oxygen consumption in the electron transport chain increases, however, the oxygen concentration in the CETs decreases. The mitochondria in the cells consume large amounts of oxygen, and oxidative phosphorylation adapts to hypoxia through the remodeling of electron transport chain [39]. NDUFA4L2 and COX4I2 can decrease mitochondrial oxygen consumption through reducing the activity of complex I and complex IV, respectively [40]. Thus, FP intervention might help prevent mitochondria from hypoxia, which keeps the energy expenditure steady.

As described above, the *Adipoq* gene of CETs that encodes adiponectin was upregulated. Playing an important role in the regulation of glucose and lipid metabolism, adiponectin increases insulin sensitivity and improves systemic lipid metabolism [41]. The decrease in adiponectin level in circulation in cases of obesity is widely related to various obesity-related diseases [42]. In previous research, we found that FP intervention significantly upregulated adiponectin, probably via the gut–brain axis [13]. The gut–brain axis also plays a key role in the control of energy balance [43]. Thus, the regulation of energy metabolism and lipid metabolism under FP intervention might be achieved through the gut–brain axis.

The relationship between dietary fiber and gene expression of the intestinal tract has been reported. For example, it was found that polyglucose regulated intestinal gene expression, upregulating an important regulator of triglyceride named FXR [44]. Furthermore, inulin fiber upregulated peptide YY and proglucagon transcripts in the colon of rats [45]. However, to our knowledge, whether the expression of colonic genes affected by FP is related to lipid metabolism and energy metabolism has not been studied. In this research, potential mechanisms are proposed, and genes that promote FA oxidation are clustered in the PPAR-signaling pathway, and genes that promote further energy metabolism are clustered in the energy-metabolism pathway.

Interestingly, there were only 28 DEGs between Group Con and Group HFD. Diet is able to modulate gene expression [46]. After the model of obese rats was successfully established by feeding rats with a high-fat diet, the gene expression of obese rats in Group HFD changed. This is the reason why they continued to develop and maintain the obesity phenotype after stopping the high-fat diet. After the obesity model was successfully established, Group Con and Group HFD were fed the same control diet, which made the gene expression of the two groups close.

The limitation of this study is that the regulation of genes related to lipid metabolism and energy metabolism in CETs by FP cannot be extended to the change in systemic metabolism. However, it provides insights into the gene expression changes of CETs associated with FP intervention and helps to understand the interaction between obesity, CETs and FP.

## 5. Conclusions

FP intervention altered the gene expression in CETs, regulating lipid metabolism and energy metabolism. By altering the gene expression of the PPAR-signaling pathway, FP intervention might accelerate FA catabolism and reduce inflammation. By regulating energy metabolism, FP intervention might facilitate the transformation of FA oxidation products to ATP. As shown here, this study offers novel hypotheses that the anti-obesity effect of FP might be closely related to the regulation of some key genes in CETs, which play an important role in lipid metabolism and energy metabolism.

## Figures and Tables

**Figure 1 foods-11-01991-f001:**
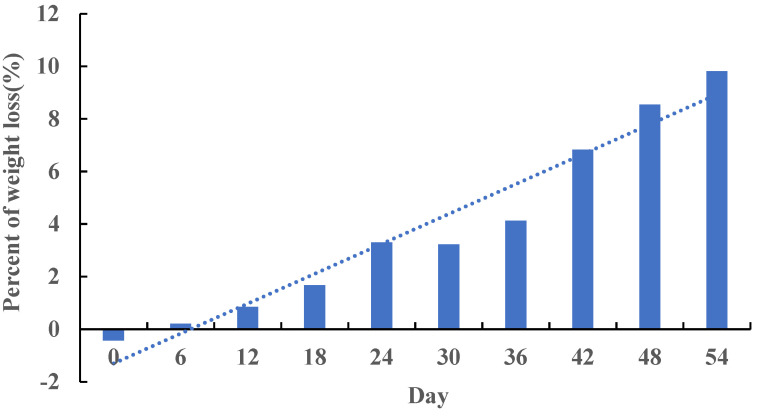
Percentage of weight loss in Group FPD compared with Group HFD. In Group FPD, rats were fed D12492 during the establishment of the obesity model and then fed an FP-containing diet; in Group HFD, rats were fed D12492 during the establishment of the obesity model and then fed AIN-93M.

**Figure 2 foods-11-01991-f002:**
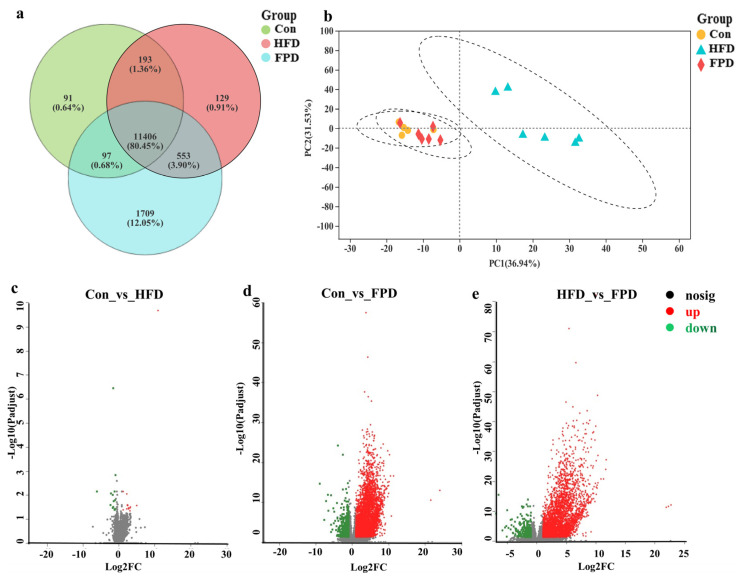
FP changed the profile of gene expression in CETs. (**a**) Venn diagram of genes; (**b**) Principal Component Analysis of genes; (**c**) DEGs between Group Con and Group HFD; (**d**) DEGs between Group Con and Group FPD; (**e**) DEGs between Group HFD and Group FPD. CETs, colonic epithelial tissues; DEGs, differentially expressed genes; Con, Group Con, rats were fed D12450B during the establishment of the obesity model and then fed AIN-93M; FPD, Group FPD, rats were fed D12492 during the establishment of the obesity model and then fed an FP-containing diet; HFD, Group HFD, rats were fed D12492 during the establishment of the obesity model and then fed AIN-93M.

**Figure 3 foods-11-01991-f003:**
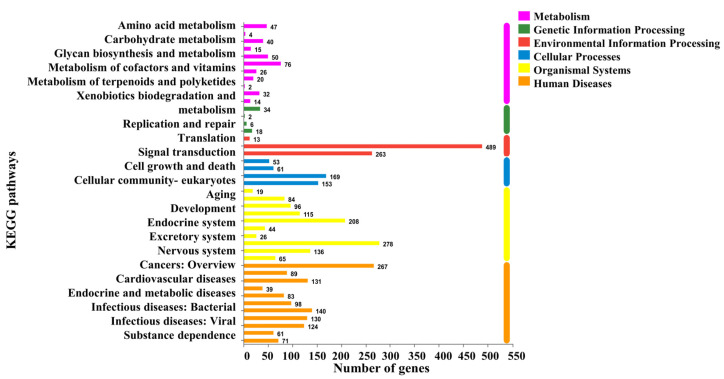
Changes in intestinal functions between Group HFD and Group FPD. HFD, Group HFD, rats were fed D12492 during the establishment of the obesity model and then fed AIN-93M; FPD, Group FPD, rats were fed D12492 during the establishment of the obesity model and then fed an FP-containing diet.

**Figure 4 foods-11-01991-f004:**
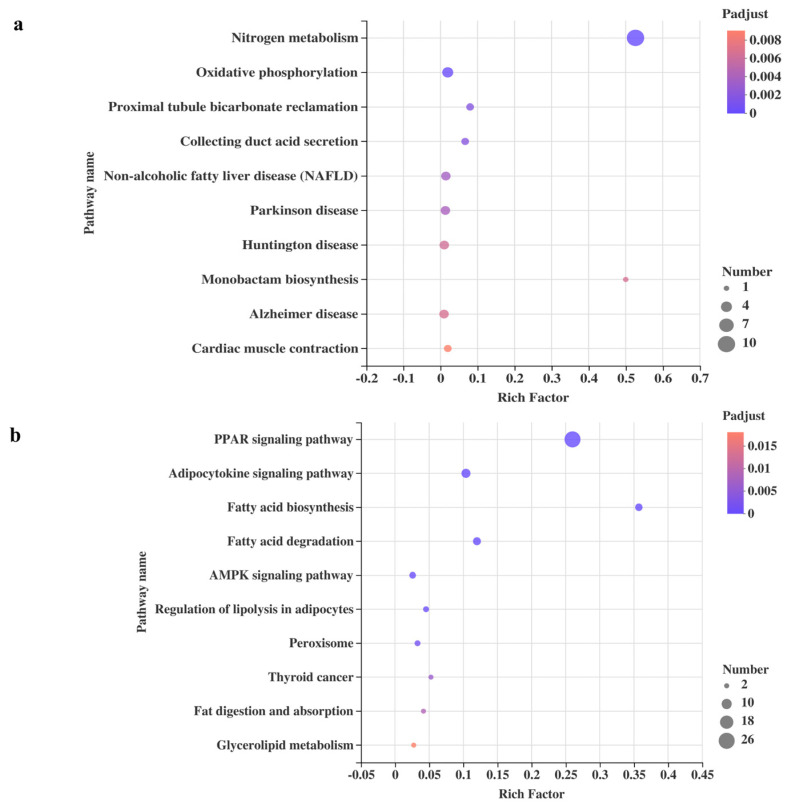
Changes in intestinal functions between Group HFD and Group FPD. (**a**) the top 10 ranked KEGG pathways of DEGs in ‘energy metabolism’; (**b**) the top 10 ranked KEGG pathways of DEGs in the PPAR-signaling pathway. HFD, Group HFD, rats were fed D12492 during the establishment of the obesity model and then fed AIN-93M; FPD, Group FPD, rats were fed D12492 during the establishment of the obesity model and then fed an FP-containing diet.

**Figure 5 foods-11-01991-f005:**
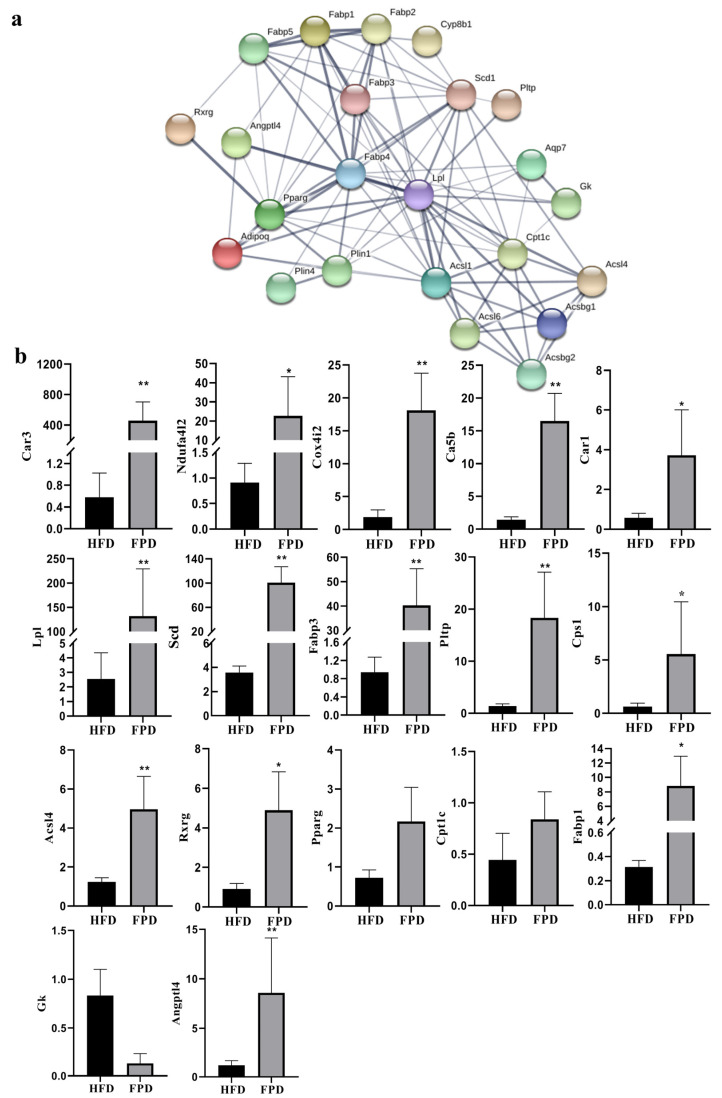
PPI network and qRT-PCR verification of DEGs. (**a**) PPI network of 23 DEGs in the PPAR-signaling pathway. The nodes represent the proteins (genes), and the edges represent the interactions of proteins; (**b**) qRT-PCR verification of DEGs in the PPAR-signaling pathway and energy-metabolism pathway. ** means *p* < 0.01. * means *p* < 0.05. DEGs, differentially expressed genes; HFD, Group HFD, rats were fed D12492 during the establishment of the obesity model and then fed AIN-93M; FPD, Group FPD, rats were fed D12492 during the establishment of the obesity model and then fed an FP-containing diet.

**Figure 6 foods-11-01991-f006:**
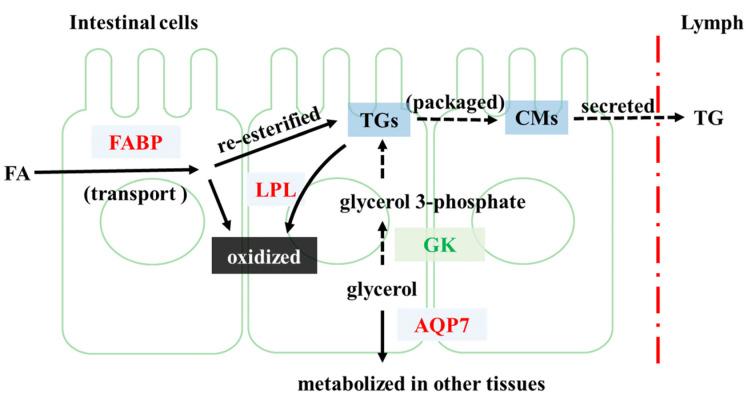
Potential regulatory mechanism by which FP may alter genes related to lipid metabolism in CETs. FP intervention upregulates FABP and LPL, promoting the transport of FA in CETs and re-oxidation and decomposition of the re-esterified TGs, which might help to reduce the secretion of TG to Lymph. GK is downregulated, reducing the conversion of glycerol to TG synthesis substrate. AQP7 is upregulated, promoting the transfer and metabolism of glycerol to other tissues. Note that proteins in red represent upregulation in group FPD compared with group HFD, while those in green indicate downregulation.

**Figure 7 foods-11-01991-f007:**
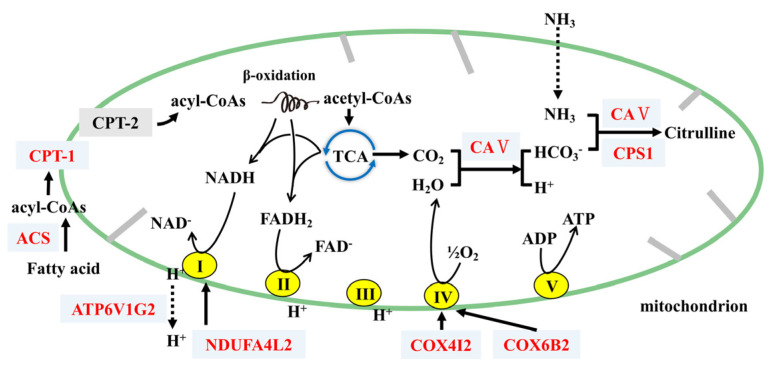
Potential regulatory mechanism by which FP may alter genes related to mitochondrial energy metabolism in CETs. FP upregulated ACS and CPT-1, promoting the conversion of FA to acyl-CoA and the transfer into mitochondria for β-oxidation and the TCA cycle where CO_2_, NADH and FADH2 are produced. Related proteins of complexes I and IV are upregulated, promoting oxidative phosphorylation and generated H_2_O. CA V catalyzes H_2_O and CO_2_ into HCO_3_^−^, and then co-catalyzes for the conversion of NH_3_ and HCO_3_^−^ to Citrulline with CPS1. Note that proteins in red represent upregulation in group FPD compared with group HFD, while those in black indicate no significant changes.

**Table 1 foods-11-01991-t001:** Detailed recipes of each diet (%).

Ingredients	Standard Diet (D12450B)	High-Fat Diet (D12492)	Control Diet (AIN-93M)	Flaxseed Polysaccharide Diet
flaxseed polysaccharide	0.00	0.00	0.00	10.00
corn starch	33.00	0.00	46.57	36.57
dextrin	3.35	16.35	15.50	15.50
casein	19.13	26.17	14.00	14.00
sucrose	34.47	9.00	10.00	10.00
cellulose	4.78	6.54	5.00	5.00
soybean oil	2.39	3.27	4.00	4.00
lard	1.91	32.06	0.00	0.00
mineral mix ain-93	3.35	4.58	3.50	3.50
vitamin mix ain-93	0.96	1.31	1.00	1.00
L-cystine	0.29	0.39	0.18	0.18
choline bitartrate	0.24	0.33	0.25	0.25

**Table 2 foods-11-01991-t002:** Sequences of the PCR primers used in gene expression studies.

Gene	Accession Number	F/P	Sequence (5′-3′)	Product Length
*β-actin*	NM_031144.3	Forward Primer	ATTGCTGACAGGATGCAGAA	109 bp
		Reverse Primer	TAGAGCCACCAATCCACACAG	
*Pparg*	NM_013124.3	Forward Primer	GTCTCACAATGCCATCAGGT	87 bp
		Reverse Primer	AGCTGGTCGATATCACTGGA	
*Fabp1*	NM_012556.2	Forward Primer	CCAAGAGAACTTTGAGCCCT	151 bp
		Reverse Primer	ATTGTGGATCACCTTGGACC	
*Rxrg*	NM_031765.1	Forward Primer	ATCCCAGCTACACAGATACCC	111 bp
		Reverse Primer	GACCCATGGCAGAAGTGATG	
*Cpt1c*	NM_001034925.2	Forward Primer	GTCTTCACTCAGTTCCGACG	117 bp
		Reverse Primer	AAGTCATTCCAGACACGCC	
*Pltp*	NM_001168543.1	Forward Primer	TTGTACCATCAAGCCGTCAG	89 bp
		Reverse Primer	GCTCGACTTCAGGCATTGTA	
*Ca5b*	NM_001005551.2	Forward Primer	CTTGAACAGTTTCGGACCCT	103 bp
		Reverse Primer	GGACAGTGCGATTCATCAGA	
*Cps1*	NM_017072.2	Forward Primer	AGCCGTTGTTTGGAATCAGT	80 bp
		Reverse Primer	GGCCATGGACATCTTGTAGG	
*Scd*	NM_139192.2	Forward Primer	CGTCAGCACCTTCTTGAGATA	168 bp
		Reverse Primer	GCGTGATGGTAGTTGTGGAA	
*Angptl4*	NM_199115.2	Forward Primer	CCACCAATGTTTCCCCCAAT	92 bp
		Reverse Primer	GCTCTTGGCACAGTTAAGGT	
*Fabp3*	NM_024162.2	Forward Primer	GCTGGGAGTAGAGTTTGACG	139 bp
		Reverse Primer	CCCATCACTTAGTTCCCGTG	
*Car1*	NM_001107660.1	Forward Primer	GTCCTGACCAATGGAGCAAA	92 bp
		Reverse Primer	GGAATCATGTTTGGCTTCGC	
*Gk*	NM_024381.2	Forward Primer	ATGGCCTAATGAAAGCTGGG	222 bp
		Reverse Primer	CAGGTTTGTCTCTGCCAAGT	
*Ndufa4l2*	NM_001271272.1	Forward Primer	ATGGCAGGAACCAGTCTAGG	79 bp
		Reverse Primer	TGAAGCCGATCATTGGGATG	
*Cox4i2*	NM_053472.1	Forward Primer	TTCGCAGAGATGAACCATCG	87 bp
		Reverse Primer	AATCACCAGAGCCGTGAATC	
*Lpl*	NM_012598.2	Forward Primer	TTGGCTCCAGAGTTTGAC	145 bp
		Reverse Primer	TGCTAATCCAGGAATCAGA	
*Acsl4*	NM_053623.1	Forward Primer	CCGAGTGAATAACTTTGGA	200 bp
		Reverse Primer	AGGAAGCCTCAGACTCAT	
*Car3*	NM_019292.4	Forward Primer	GCTCTGCTAAGACCATCC	117 bp
		Reverse Primer	ATTGGCGAAGTCGGTAGG	

**Table 3 foods-11-01991-t003:** Information of three KEGG pathways and DEGs enriched.

KEGG Pathway	Gene Name	Gene Description	Regulation
Oxidative phosphorylation	*Ndufa4l2*	NDUFA4, mitochondrial complex associated like 2	Upregulated
	*Atp6v1g2*	ATPase H+ transporting V1 subunit G2	Upregulated
	*Cox4i2*	cytochrome c oxidase subunit 4i2	Upregulated
	*Cox6b2*	cytochrome c oxidase subunit VI b polypeptide 2	Upregulated
Nitrogen metabolism	*Cps1*	carbamoyl-phosphate synthase 1	Upregulated
	*Car3*	carbonic anhydrase 3	Upregulated
	*Car1*	carbonic anhydrase I	Upregulated
	*Ca5b*	carbonic anhydrase 5B	Upregulated
	*Car8*	carbonic anhydrase 8	Upregulated
	*Car13*	carbonic anhydrase 13	Upregulated
	*Car2*	carbonic anhydrase 2	Upregulated
	*Car12*	carbonic anhydrase 12	Upregulated
	*Car14*	carbonic anhydrase 14	Upregulated
	*Car4*	carbonic anhydrase 4	Upregulated
PPAR-signaling pathway	*Plin4*	perilipin 4	Upregulated
	*Fabp1*	fatty acid binding protein 1	Upregulated
	*Fabp3*	fatty acid binding protein 3	Upregulated
	*Cyp8b1*	cytochrome P450, family 8, subfamily b, polypeptide 1	Upregulated
	*Scd*	stearoyl-CoA desaturase	Upregulated
	*Fabp4*	fatty acid binding protein 4	Upregulated
	*Lpl*	lipoprotein lipase	Upregulated
	*Acsl6*	acyl-CoA synthetase long-chain family member 6	Upregulated
	*Plin1*	perilipin 1	Upregulated
	*Pltp*	phospholipid transfer protein	Upregulated
	*Angptl4*	angiopoietin-like 4	Upregulated
	*Rxrg*	retinoid X receptor gamma	Upregulated
	*Adipoq*	adiponectin, C1Q and collagen domain containing	Upregulated
	*Acsbg2*	acyl-CoA synthetase bubblegum family member 2	Upregulated
	*Fabp5*	fatty acid binding protein 5, epidermal	Upregulated
	*Fabp2*	fatty acid binding protein 2	Upregulated
	*Aqp7*	aquaporin 7	Upregulated
	*Cpt1c*	carnitine palmitoyl transferase 1c	Upregulated
	*Acsl4*	acyl-CoA synthetase long-chain family member 4	Upregulated
	*Pparg*	peroxisome proliferator-activated receptor gamma	Upregulated

## Data Availability

Not applicable.

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
