# Peer review of "Flaxseed Polysaccharide Alters Colonic Gene Expression of Lipid Metabolism and Energy Metabolism in Obese Rats"

_foods, 2022, doi:10.3390/foods11131991_

Round 1

Reviewer 1 Report

  • A brief summary 

Since humanity faces nowadays a major obesity epidemic, it is of great importance to test various solutions to this challenge. The researchers aim to prove the anti-obesity effect of flaxseed polysaccharides, used as dietary fiber, by the transformation of fatty acid oxidation products to ATP. A flaxseed polysaccharide diet regulated the expression of lipid metabolism (by accelerating fatty acids catabolism) and energy metabolism-related genes of colonic epithelial tissues. The study offers novel hypotheses that the anti-obesity effect of FP might be closely related to the regulation of some key genes in CETs. The article is rich in data (both tables and figures) which is explicit and coherent.

  • General concept comments
    Article: highlighting areas of weakness, the testability of the hypothesis, methodological inaccuracies, missing controls, etc.

The manuscript is relevant to the field and is written in a well-structured manner. The references are mostly recent publications and don’t include an excessive number of self-citations. Some of the figures are too small, even with zooming in, it can be a bit difficult to read and the text doesn’t discuss all the data present in the figures.

The study has a limiting factor which is the small sample size. This weakness can influence the results and the statistical interpretation, by reducing the power of the study and increasing the margin of error. Compared to other research in the field, it would have been preferable to have a higher sample size and several repetitions per group.

  • Specific comments 

Row 10 – please correct “polysaccharide” in polysaccharides (plural) throughout the article

Row 12 – please replace in with on in “study was performed on”

Row 15 – please correct KEGG pathways were found, not “found out”

Row 26 – please add a comma after behavioral

Row 26 – please format de citation Hill, 2006 according to the Journal and add it to References

Row 32 – please add induced “by” a high-fat diet

Row 38 – As dietary fibers (add an s), flaxseed polysaccharides “are”

Row 42 – add a comma after satiety

Row 49 – please add a dot at the end of the sentence

Row 54 – “Professor Yong Wang in Jinan University”, replace “in” with” at”

Row 56 – add a comma after precipitation

Row 62 – please add eighteen male Sprague-Dawley “rats”

Row 64 –  delete the dot after China, in the brackets

Row 64 – with “a” standard diet

Row 65 and 66 – ”the” control group and ”the” obesity group

Row 66 – rats “were” fed, replace was with were

Row 66 – suggestion of correcting “were fed with a standard diet (D12450B), and rats in the obesity group were given a high-fat diet (D12492)”

Row 70 – rats in “the” obesity group, please correct n=6, by deleting the spaces

Row 71 – please be consistent and write Group FPD with a capital G (as for Group HFD)

Row 74 – “are shown”  in Table 1

Row 76 -  after “the” execution

Row 140 – “are shown” in Figure 4a

Row 144 – Figure 4b ”shows”

Row 164 – please delete Venn, as it appears twice

Row 206 and in the whole article – please edit chemical formulas accordingly (with subscripts like CO2, H2O, etc.)

Row 220 – fatty acids, plural

Row 230 – in the synthesis

Row 237 – heterodimeric – 1 word

Row 249 – add a dot at the end of the sentence

Row 257 – please use chemical formulas with subscripts

Row 260 – increase, not increased

Row 276 – prevent, keeps

Row 278 – encodes

Row 279 – research (singular)

Row 295 – please delete “had” and leave only changed

Row 321 – please delete the extra dot at the end of the sentence, after research.

Author Response

  1. Englinsh language and style have been checked.
  2. We have replaced all figures with large size.
  3. The reviewer gave a very good suggestion about sample size. We adopted 6 rats based on many publised literatures. In the future research, we will consider the number of experimental animals according to the reviewer's suggestion. 

Reviewer 2 Report

Regarding the manuscript entitled '' Flaxseed polysaccharide altering colonic gene expression of lipid metabolism and energy metabolism in obese rat ''

Abstract.

The abstract is poorly written and needs revision.

Why energy is started with capital letter, please revise

Please define the most important genes in these pathways. Why the authors mentioned

hypothesis in the abstract. The abstract should define the main results or findings.

One sentence should be added at the end of abstract to conclude the findings of the study and recommendation.

Introduction is very short and needs more information about the objectives, hypothesis, and brief description of the flaxseed polysaccharide's mode of action. The authors should focus on the novelty of the study in more details.

L26.  (Hill, 2006), revise what is the number, it is not in the reference list

L34. Induced, please be precise

Materials and Methods

L52, 53. Name of the company should in parenthesis (DBI Bioscience co., Shanghai, China). Please revise others

L56. Add ref.

L66. Why the number of animals in the control is differ than in treatment?

L68. Please add average weight of each group in parenthesis with SE or SEM

L100. Do you think one internal control gene is enough for accurate results?

L133. And so on!!

Figure 1 is difficult to understand; I see only one column.

Table 1 should be presented before figure 1

Figure 2 is poorly presented, please add more clear view

Table 1 and Table 2 should be in materials and methods section

All the figures should be revise by the authors to be clear for the readers. The font's size is very small to read, and the resolution is not good. I strongly recommended the authors to revise figures.

L240. Please define genes related to lipid metabolism, that changed in this study.

L278. Adiponectin play important role in insulin sensitivity and obesity, please add something related to this part.  

Author Response

Response to the review's comments:

  1.  

The abstract is poorly written and needs revision.

Why energy is started with capital letter, please revise.

Please define the most important genes in these pathways. Why the authors mentioned hypothesis in the abstract. The abstract should define the main results or findings.

One sentence should be added at the end of abstract to conclude the findings of the study and recommendation.

Response: Thanks. We modified the abstract. Energy is modified to start with a lowercase letter. We deleted the hypothesis. At the end of abstract, one sentence was used to conclude the findings of the study and recommendation.

  1. Introduction is very short and needs more information about the objectives, hypothesis, and brief description of the flaxseed polysaccharide's mode of action. The authors should focus on the novelty of the study in more details.

Response: Thanks very much. We supplemented more information about the objectives, hypothesis, and brief description of the flaxseed polysaccharide's mode of action in the manuscript and focus on the novelty of the study in more details.

  1. (Hill, 2006), revise what is the number, it is not in the reference list.

Response: Thanks. We added this reference as number 1 in the reference list.

  1. Induced, please be precise.

Response: Thanks.“Induced”was replaced by “influenced”.

  1. L52, 53. Name of the company should in parenthesis (DBI Bioscience co., Shanghai, China). Please revise others

Response: Thanks. We revise all of the name of the company in parenthesis.

  1. Add ref.

Response: Thanks. Reference had been added.

  1. Why the number of animals in the control is differ than in treatment?

Response: Thanks. As showed in the article, the rats were divided into control group ( n=6) and obesity group (n=12). After the obesity model was established, rats in obesity group were randomly divided (n = 6 for each) into High-fat group and FP-diet group, which were fed with high-fat diet and FP diet respectively. Therefore, the number of animals in the control is differ than in treatment.

  1. Please add average weight of each group in parenthesis with SE or SEM.

Response: Thanks. Average weight of each group was added.

  1. Do you think one internal control gene is enough for accurate results?

Response: Thanks. From our previous research experience on transcriptome genes in colon epithelial tissue, we can see that when β-actin is used as an internal control gene, it can get relatively stable results. In order to maintain the consistency of research results, it is very necessary to use the same internal control gene. Limited by the cost and sample size, we think that selecting a relatively stable internal control gene is enough to obtain accurate results.

  1. And so on!!

Response: Thanks. We deleted “and so on”.

  1. Figure 1 is difficult to understand; I see only one column.

Response: Thanks. We replaced the original figure with a clearer one. The columns represented percentages of weight loss in Group FPD compared with Group HFD in different days. Since the proportion change is calculated based on the average weight of each group, it can only be expressed in multiple columns.

  1. Table 1 should be presented before figure 1

Response: Thanks. We showed table 1 before figure 1.

  1. Figure 2 is poorly presented, please add more clear view

Response: Thanks. We added more clear view of Figure 2.

  1. Table 1 and Table 2 should be in materials and methods section

Response: Thanks. We put Tables 1 and 2 in the materials and methods section.

  1. All the figures should be revise by the authors to be clear for the readers. The font's size is very small to read, and the resolution is not good. I strongly recommended the authors to revise figures.

Response: Thanks. Following the suggestions of the reviewers, we made major adjustments to the pictures, including font, size and resolution.

  1. Please define genes related to lipid metabolism, that changed in this study.

Response: Thanks. We have defined genes related to lipid metabolism, that changed in this study.

  1. Adiponectin play important role in insulin sensitivity and obesity, please add something related to this part.

Response: Thanks. We have added the following sentence to this section: “Playing an important role in the regulation of glucose and lipid metabolism, adiponectin increases insulin sensitivity and improves systemic lipid metabolism[1].The decrease of adiponectin level in circulation under obesity is widely related to various obesity related diseases[2].”

  • Adiyaman, S. C.; Ozer, M.; Saydam, B. O.; Akinci, B. The Role of Adiponectin in Maintaining Metabolic Homeostasis. Current diabetes reviews 2020,16(2), 95–103.
  • Ohashi, K.; Yuasa, D.; Shibata, R.; Murohara, T.; Ouchi, N. Adiponectin as a Target in Obesity-related Inflammatory State. Endocrine, metabolic & immune disorders drug targets 2015, 15(2), 145–150.

Round 2

Reviewer 2 Report

Abstract

Dear authors, thank you for the revisions. The manuscript was improved. However, I still have some minor comments.

Still, I am not happy with the abstract. The abstract is weak and not strong enough for papers published in Foods.

L70-71. Add units (g)

Table 2. please add accession number and nucleotide size

Author Response

Dear Sir/Madam,

     Thank you for all suggestions. We have revised each point according to your advice.

  1. We have rewritten the abstract.
  2. 'g' has been added.
  3.  The accession number and nucleotide size have been supplemented into Table 2. 

       Thank you very much!   Best  regards,

                                                                   Xichun Peng
